# Daphnetin Protects Schwann Cells Against High-Glucose-Induced Oxidative Injury by Modulating the Nuclear Factor Erythroid 2-Related Factor 2/Glutamate–Cysteine Ligase Catalytic Subunit Signaling Pathway

**DOI:** 10.3390/plants13213066

**Published:** 2024-10-31

**Authors:** Chih-Yuan Ko, Run-Tian Meng, Chung-Hsin Wu, Thi Kim Ngan Nguyen, Yu-En Chen, James Swi-Bea Wu, Wen-Chung Huang, Szu-Chuan Shen

**Affiliations:** 1Department of Clinical Nutrition, The Second Affiliated Hospital of Fujian Medical University, Quanzhou 362000, China; 2School of Public Health, Fujian Medical University, Fuzhou 350122, China; 3School of Life Science, National Taiwan Normal University, Taipei 10617, Taiwan; 4Graduate Program of Nutrition Science, National Taiwan Normal University, Taipei 11677, Taiwan; 5Graduate Institute of Food Science and Technology, National Taiwan University, Taipei 10617, Taiwan; 6Graduate Institute of Health Industry Technology, Chang Gung University of Science and Technology, Taoyuan 33303, Taiwan

**Keywords:** diabetic peripheral neuropathy, Schwann cells, *Ficus formosana* Maxim., daphnetin, Nrf2, RSC96 cells, Bcl-2, Bax, apoptosis

## Abstract

Diabetic peripheral neuropathy (DPN), a common complication of diabetes mellitus, is primarily characterized by damage to Schwann cells caused by oxidative stress under hyperglycemic conditions. Recently, we demonstrated the ability of coumarin-rich *Ficus formosana* Maxim. to alleviate DPN in ovariectomized diabetic mice. However, the underlying mechanisms remain unclear. In this study, we established an in vitro DPN model using RSC96 Schwann cells exposed to high glucose levels. Daphnetin, a natural coumarin found abundantly in *Ficus formosana* Maxim., was co-incubated with Schwann cells in a high-glucose medium to investigate its protective effects against DPN. The free radical scavenging capacity of daphnetin was evaluated, along with assessments of cell viability, apoptosis, H_2_O_2_ levels, and the expression of proteins by the nuclear factor erythroid 2-related factor 2 (Nrf2)/glutamate–cysteine ligase catalytic subunit (GCLC) pathway in RSC96 Schwann cells. The results showed that daphnetin was non-toxic within the tested concentration range of 6.25 μM to 50 μM in RSC96 Schwann cells. Moreover, daphnetin significantly improved cell viability, exhibited strong antioxidant activity, reduced H_2_O_2_ levels, and regulated the Nrf2/GCLC pathway protein expressions in RSC96 cells cultured in high-glucose medium. Additionally, daphnetin influenced apoptosis-related proteins by decreasing the expression levels of Bax and Caspase 3, while increasing the Bcl-2 expression level in high-glucose-treated RSC96 cells. These findings suggest that daphnetin may alleviate oxidative stress induced by high glucose levels through activation of the Nrf2/GCLC pathway and inhibition of Schwann cell apoptosis, underscoring its potential as a therapeutic agent for DPN.

## 1. Introduction

Diabetes mellitus (DM) is a chronic metabolic disorder characterized by hyperglycemia and is associated with numerous complications. Diabetic peripheral neuropathy (DPN) is one of the most common complications, affecting approximately half of all diabetic patients worldwide [1]. DPN is a debilitating condition that results in sensory loss, pain, and a higher risk of foot ulcers and amputations [1]. Schwann cells, the main glial cells of the peripheral nervous system, play a crucial role in nerve regeneration and repair by maintaining axonal integrity and promoting myelination [2,3]. Under hyperglycemic conditions, Schwann cells are highly vulnerable to oxidative stress, leading to apoptosis and impaired nerve function [2,4].

Oxidative stress, driven by the overproduction of reactive oxygen species (ROS), is a key contributor to the development of DPN [4]. The cellular imbalance between ROS and the antioxidant defenses results in mitochondrial dysfunction, lipid peroxidation, and ultimately Schwann cell death [1,5,6]. The nuclear factor erythroid 2-related factor 2 (Nrf2) pathway plays a crucial role in regulating the cellular antioxidant defense system [7]. When activated, Nrf2 enhances the expression of several antioxidant enzymes, including the glutamate–cysteine ligase catalytic subunit (GCLC), which protect Schwann cells from oxidative damage [8,9,10].

In our previous research, we demonstrated that coumarin-rich *Ficus formosana* Maxim., a traditional medicine used in the treatment of DM in the Orient, alleviated peripheral neuropathy by reducing oxidative damage in ovariectomized diabetic mice [6]. Coumarin is a group of natural compounds abundant in plants, known for their various biological activities, including antioxidant [11], anti-inflammatory [12], and anticancer properties [13]. Daphnetin (Figure 1), a coumarin derivative, has been shown to modulate oxidative stress responses in cell models [10]. Additionally, daphnetin has been reported to activate Nrf2 expression and its downstream antioxidant targets, providing neuroprotective effects against oxidative stress in vitro [8,10]. Given the severe side effects of conventional drugs, there is an urgent need to explore natural therapeutic strategies for DM and its complications, such as DPN. However, the potential of daphnetin to prevent DPN remains inconclusive. This study aims to investigate the therapeutic potential of daphnetin in protecting Schwann cells from high-glucose-induced oxidative injury by modulating the Nrf2/GCLC signaling pathway.

## 2. Results

### 2.1. Effect of Daphnetin on RSC96 Schwann Cells’ Viability Under High-Glucose Conditions

The results demonstrated that the maximum non-toxic concentration of daphnetin for RSC96 Schwann cells was 50 μM (Figure 2), which was subsequently used in further experiments. Increasing glucose concentrations resulted in a progressive reduction in cell viability. This concentration was then used in further experiments in this study. Additionally, increasing glucose concentrations resulted in a progressive decrease in cell viability. Various glucose concentrations (5.5 mM, 50 mM, 100 mM, 125 mM, and 150 mM) were used to simulate both normal and hyperglycemic conditions in RSC96 Schwann cells. Cell viability decreased by 17% at 150 mM (high glucose, HG group) compared to 5.5 mM (normal glucose, NG group) (Figure 3), and thus 150 mM glucose was used in subsequent experiments.

Figure 4 shows that, compared to the NG group, the viability of RSC96 cells in the HG group was significantly reduced, suggesting that a high-glucose environment could damage Schwann cells and even cause their death. However, daphnetin significantly increased cell viability by 7% compared to the HG group, indicating that daphnetin may reduce Schwann cell damage in a high-glucose environment.

### 2.2. Free Radical Scavenging Ability of Daphnetin

The 2,2-Diphenyl-1-picrylhydrazyl (DPPH) assay was used to evaluate the hydrogen-donating ability of the antioxidants, and daphnetin had an IC_50_ value of 435.4 mM for scavenging DPPH free radicals (Figure 5A). This indicates that daphnetin has a strong hydrogen proton-donating ability. In addition, the ABTS assay showed an IC50 of 146.1 mM for daphnetin (Figure 5B), indicating its excellent antioxidant capacity.

### 2.3. Impact of Daphnetin on Hydrogen Peroxide (H_2_O_2_) Content of RSC96 Schwann Cells Under High-Glucose Conditions

Generally, 2′, 7′-dichlorodihydrofluorescein diacetate (DCFH-DA) is used to measure the H_2_O_2_ content in “live” cells at different time points in order to avoid errors caused by apoptotic cells. In this study, the highest concentration of H_2_O_2_ in Schwann cells in a high-glucose environment occurred during the sixth hour. Figure 6 shows that high glucose levels increase oxidative stress in Schwann cells, as evidenced by the higher H_2_O_2_ content in the HG group when compared to the NG group. In contrast, daphnetin at a concentration of 25 μM reduced H_2_O_2_ levels by around 20% compared to the HG group (Figure 6A,B), demonstrating its inhibiting ability on H_2_O_2_ production in RSC96 Schwann cells in a high-glucose medium.

### 2.4. Regulation of Nrf2 Expression in RSC96 Schwann Cells by Daphnetin Under High-Glucose Conditions

The Nrf2 mRNA levels of RSC96 cells cultured with daphnetin in high-glucose conditions are shown in Figure 6A. The results revealed that 25 μM and 50 μM daphnetin significantly increased Nrf2 mRNA levels by 96% and 94%, respectively, compared to the HG group (Figure 7A). Moreover, it was observed that 50 μM daphnetin significantly increased nuclear Nrf2 expression by 6-fold compared to the HG group (Figure 7B).

### 2.5. Regulation of Antioxidant Proteins and Glutathione (GSH) Levels in RSC96 Schwann Cells by Daphnetin Under High-Glucose Conditions

The protein expression of GCLC and heme oxygenase-1 (HO-1) in RSC96 Schwann cells was determined by co-culture with daphnetin in a high-glucose medium. The results showed that 25 μM and 50 μM daphnetin increased GCLC expression by 84% and 38% (Figure 8A), respectively; however, no significant difference was observed for HO-1compared to the HG group (Figure 8B). In addition, 25 μM and 50 μM daphnetin significantly increased GSH levels compared to the HG group (Figure 8C).

### 2.6. Regulation of Apoptosis of RSC96 Schwann Cells by Daphnetin Under High-Glucose Conditions

Figure 9A,B shows that the HG group significantly increased apoptosis 2.7-fold; however, 12.5 μM, 25 μM, and 50 μM daphnetin significantly reduced cell apoptosis by 60–80% compared to the NG group. In addition, the protein expression of Bax and Caspase 3 in the HG group was found to be approximately 50% higher than that in the NG group (Figure 9C,D). However, 6.25–50 μM and 6.25–25 μM daphnetin significantly inhibited Bax expression by 10–40% and Caspase 3 expression by 25–35%, respectively, in RSC96 Schwann cells compare to the HG group (Figure 9C,D). Moreover, daphnetin also significantly increased the level of anti-apoptotic protein Bcl-2 protein expression in Schwann cells compared to the HG group (Figure 9E). These results suggested that daphnetin inhibits the pro-apoptotic proteins Bax and Caspase 3, and increases the anti-apoptotic protein Bcl-2 in high-glucose-induced apoptosis in Schwann cells.

### 2.7. Daphnetin-Induced Morphological Alterations of RSC96 Schwann Cells in High-Glucose Environments

In this study, the RSC96 Schwann cells remained oval-shaped in the HG group, indicating that a high-glucose medium impairs the differentiation of cells (Figure 10B). However, a spindle-shaped or bipolar-shaped cell morphology was observed after co-culturing with daphnetin in a high-glucose medium (Figure 10C–F), suggesting that daphnetin promoted Schwann cells differentiation in a high-glucose environment.

## 3. Discussion

Our study demonstrates that daphnetin significantly protects Schwann cells from high glucose-induced oxidative stress and apoptosis by modulating the Nrf2/GCLC signaling pathway. These findings are consistent with previous research, which underscores the role of oxidative stress in DPN and the critical function of the Nrf2 pathway in reducing oxidative damage.

Previous studies have shown that the progression of DPN is associated with poor blood glucose control and a long duration of diabetes [14]. In patients with type 2 DM, the prevalence of DPN exceeds 19% after 5–10 years of disease [14]. Excess oxidative stress caused by high blood glucose levels plays a key role in the development of DPN [4]. Phytochemicals from plants, including vegetables and herbs, have been identified for their strong antioxidant properties [11]. Coumarin, recognized for its distinctive aroma, is widely used as a food flavoring agent following its synthesis [15]. In humans, coumarin is metabolized by the enzyme 7-hydroxylase in liver microsomes into 7-hydroxycoumarin, which is then conjugated and excreted in urine [16].

This study investigated the effects of daphnetin, a natural coumarin derivative, on the viability of RSC96 Schwann cells under high-glucose conditions. While standard culture medium contains approximately 5.5 mM glucose [17], we observed that cell viability decreased as glucose concentrations increased, consistent with previous findings [2]. However, daphnetin improved Schwann cell viability in these high-glucose environments. In DPN models, Schwann cells have been cultured in glucose concentrations ranging from 44.4 to 150 mM to induce diabetic neuropathy [8,17,18,19,20,21]. In the present study, we used a high glucose concentration of 150 mM to simulate DPN in Schwann cells, similar to previous studies.

Schwann cells are essential for maintaining peripheral nerve function, and oxidative stress plays a significant role in their dysfunction under hyperglycemic conditions [4]. This study also revealed that daphnetin possesses strong antioxidant properties. Both the DPPH and ABTS assays confirmed daphnetin’s potent antioxidant capacity. Notably, daphnetin effectively reduced oxidative stress markers, such as H_2_O_2_, indicating its protective effect on RSC96 Schwann cells incubated in high-glucose conditions. This antioxidant capacity may be linked to the hydroxyl groups present in the chemical structure of daphnetin [22]. Moreover, this study demonstrated that daphnetin exerts its protective effects by activating the Nrf2 pathway, which plays a crucial role in mitigating oxidative stress and promoting Schwann cell survival [9]. Nrf2 is a transcription factor that, upon translocation from the cytoplasm to the nucleus, regulates the expression of antioxidant enzymes. In this study, the increased expression of Nrf2 mRNA and elevated nuclear Nrf2 protein levels resulted in the upregulation of GCLC, promoting intracellular GSH synthesis and effectively reducing oxidative damage under high-glucose conditions [8,23]. The observed increase in GSH levels and reduction in H_2_O_2_ are consistent with findings that enhancing the antioxidant response via the Nrf2 pathway improves cell viability and reduces oxidative stress markers [24]. Additionally, the significant decrease in H_2_O_2_ levels suggests that the antioxidant effects of daphnetin are mediated through the Nrf2/GCLC pathway, with no significant changes in HO-1 expression. These findings highlight the potential of daphnetin in counteracting oxidative stress related to DPN.

Oxidative stress is known to overwhelm cellular antioxidant defenses, leading to apoptosis, a hallmark of DPN [1,5,6]. Under hyperglycemic conditions, Schwann cells are particularly vulnerable to apoptosis driven by excessive ROS production [25], with excess H_2_O_2_ contributing to the apoptosis of these cells [26]. To assess this, we used flow cytometry and Western blotting to measure the extent of apoptosis in RSC96 cells and the expression of apoptosis-related proteins, including Bax, Caspase 3, and Bcl-2. It is well established that Bcl-2 is negatively associated with apoptosis, while Bax and Caspase 3 are positively associated with it [27]. In this study, daphnetin was found to modulate the expression of these apoptosis-related proteins, reducing pro-apoptotic Bax and Caspase 3 levels while increasing the anti-apoptotic Bcl-2 levels in Schwann cells exposed to high-glucose conditions. These findings are consistent with previous research demonstrating similar protective effects against hyperglycemia-induced oxidative damage in Schwann cells [17].

Normally, RSC96 Schwann cells exhibit an oval shape before differentiation, transitioning to a spindle-shaped or bipolar morphology after differentiation [28]. High-glucose-induced damage to RSC96 Schwann cells often results in morphological changes, such as swelling and vacuolation, which lead to the destruction of organelles [29]. The results of the current study suggest that daphnetin not only inhibits apoptosis but also improves cell morphology and enhances the proliferation of Schwann cells cultured in a high-glucose environment.

This study has several limitations. In vitro experiments may not fully replicate the complexity of Schwann cells in the in vivo environment in diabetic patients. Future research should build on these findings by exploring additional cell lines to better understand cellular responses under diabetic conditions, as well as conducting in vivo studies to evaluate the therapeutic potential of daphnetin in diabetic models. Furthermore, it is essential to assess the long-term effects and safety profiles of these compounds in clinical settings if they are to be developed as therapeutic agents.

## 4. Materials and Methods

### 4.1. Cell Culture

The RSC96 rat Schwann cell line was purchased from the Food Industry Research and Development Institute (Hsinchu, Taiwan). The cells were cultured in Dulbecco’s Modified Eagle Medium (Thermo Fisher, Waltham, MA, USA) containing 10% fetal bovine serum (FBS; Wellington, CO, USA) and 1% non-essential amino acids (Beit-Haemek, Israel), and maintained at 37 °C in a 5% CO_2_ incubator.

### 4.2. Treatment and Viability Assay

RSC96 Schwann cells were incubated in media containing 5.5, 50, 100, 125, and 150 mM glucose, 0.5% DMSO medium contain 150 mM glucose with fresh prepared 6.25, 12.5, 50, 100, 125, and 150 μM daphnetin (C_9_H_6_O_4_; Purity ≥ 97%; Sigma-Aldrich, St. Louis, MO, USA), respectively, for 24 h. For the viability assay, cells were treated with 3-(4,5-dimethylthiazol-2-yl)-2,5-diphenyltetrazolium bromide (MTT; Sigma-Aldrich, St. Louis, MO, USA) for 4 h, and then analyzed spectrophotometrically at 517 nm. Viability was determined as a percentage of control cells treated with the vehicle alone.

### 4.3. DPPH Radical Scavenging Activity

The DPPH radical scavenging assay was used to evaluate the antioxidant activity of the samples [30]. A 0.1 mM solution of DPPH (Sigma-Aldrich, St. Louis, MO, USA) in methanol was prepared. Samples were mixed with the DPPH solution and incubated in the dark for 30 min. The absorbance was measured at 517 nm.

### 4.4. ABTS Radical Scavenging Activity

The ABTS radical cation decolorization assay was used to measure the ability of antioxidants to scavenge ABTS [30]. ABTS was generated by mixing 7 mM ABTS (Sigma-Aldrich, St. Louis, MO, USA) with 2.45 mM potassium persulfate and allowing the mixture to react in the dark for 12–16 h. The ABTS solution was then diluted to an absorbance of 0.7 at 734 nm. Samples were mixed with the ABTS solution, incubated for 6 min in the dark, and the absorbance was measured at 734 nm.

### 4.5. GSH Content Determination

GSH levels were measured using a colorimetric assay. Cells were treated with various concentrations of samples. After incubation, cells were lysed, and GSH levels were determined by reacting cell lysates with 5,5′-dithiobis-(2-nitrobenzoic acid) (Sigma-Aldrich, St. Louis, MO, USA) and measuring the absorbance at 412 nm.

### 4.6. H_2_O_2_ Content Determination

Intracellular H_2_O_2_ levels were measured using DCFH-DA (Sigma-Aldrich, St. Louis, MO, USA). Cells were incubated with DCFH-DA, which is converted to the fluorescent compound dichlorodihydrofluorescein in the presence of H_2_O_2_. The fluorescence intensity was measured using a flow cytometer with excitation at 485 nm and emission at 535 nm [31].

### 4.7. Cell Apoptosis Analysis

Apoptosis was assessed using the Annexin V-FITC apoptosis detection kit (Dojindo, Munich, Germany), followed by flow cytometry. Cells were stained with annexin V-FITC and propidium iodide (PI; Thermo Fisher Scientific, Waltham, MA, USA) to distinguish between live, apoptotic, and necrotic cells. The percentage of apoptotic cells was analyzed using flow cytometry.

### 4.8. Real-Time Quantitative PCR Analysis

Total RNA was extracted from cells using the TRIzol reagent (Zymo Research, Irvine, CA, USA). cDNA was synthesized using a reverse transcription kit (Protech, Taipei, Taiwan). Real-time quantitative PCR was performed using specific primers for target genes and the SYBR Green master mix. Primer sequences of Nrf2 and β-actin are (5′-GTACAACCCTTGTCACCATCTC-3′, 5′-TCCGATGACCAGGACTTACA-3′ and 5′-CCGTCTTCCCCTCCATCG-3′, 5′-GTCCCAGTTGGTGACGATGC-3′, respectively. The relative expression levels of genes were calculated using the ΔΔCt method with β-actin as the internal control.

### 4.9. Western Blot Analysis

The protein extraction methods used in this study were based on our previous protocol [32]. The protein concentration in the cell extract was determined using a Bio-Rad protein assay dye reagent (Richmond, VA, USA). Aliquots of the supernatant protein were separated by standard SDS-PAGE and then electrophoretically transferred to polyvinylidene difluoride membranes. The membranes were incubated overnight at 4 °C with the following primary antibodies: anti-Nrf2 (1:1000, Cat. No. 33649, Cell Signaling Technology), anti-caspase-3 (1:1000, Cat. No. 9662, Cell Signaling Technology, Danvers, MA, USA), anti-HO-1 (1:1000, Cat. No. 82206, Cell Signaling Technology, Danvers, MA, USA), anti-Lamin-b1 (1:1000, Cat. No. 13435, Cell Signaling Technology), anti-GCLC (1:2000, Cat. No. ab190685, Abcam, Cambridge, UK), anti-Bcl-2 (1:1000, Cat. No. ab182858, Abcam), anti-Bax (1:200, Cat. No. ab32503, Abcam), or anti-β-actin (1:1000, Cat. No. ab8226, Abcam, Cambridge, UK). Subsequently, the membranes were incubated with anti-mouse IgG or anti-rabbit IgG secondary antibodies and washed three times for 5 min each. Protein bands were detected and captured using the UVP Biospectrum imaging system (Level, Cambridge, UK). Finally, all relative protein expressions were normalized to β-actin.

### 4.10. Statistical Analysis

The experimental data were analyzed using SPSS version 22 (SPSS Inc., Chicago, IL, USA). The means and standard errors of each group were calculated, and one-way ANOVA was used to determine significant differences between groups. Post hoc tests were performed using Duncan’s multiple range test to identify significant differences. Statistical significance was defined as *p* < 0.05.

## 5. Conclusions

This study highlights the protective effects of daphnetin, a natural coumarin derivative, on RSC96 Schwann cells exposed to high-glucose conditions. Daphnetin facilitated the nuclear translocation of Nrf2, increased the expression of antioxidant proteins such as GCLC and GSH but not HO-1, and consequently reduced oxidative stress damage in the Schwann cells. Additionally, daphnetin inhibited apoptosis by regulating the expression of Bax, Caspase 3, and Bcl-2 proteins through the suppression of oxidative stress in Schwann cells (Figure 11). In summary, daphnetin may mitigate oxidative damage caused by hyperglycemia in DM and shows potential as a therapeutic option for managing DPN.

## Figures and Tables

**Figure 1 plants-13-03066-f001:**
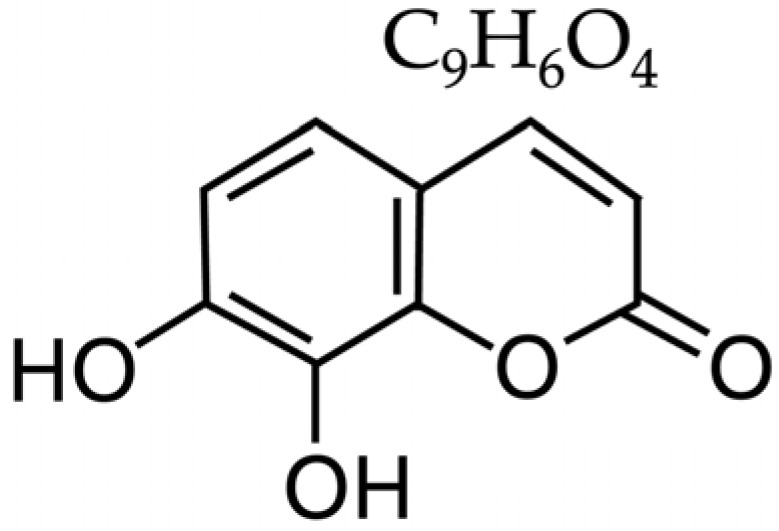
The chemical structure of daphnetin.

**Figure 2 plants-13-03066-f002:**
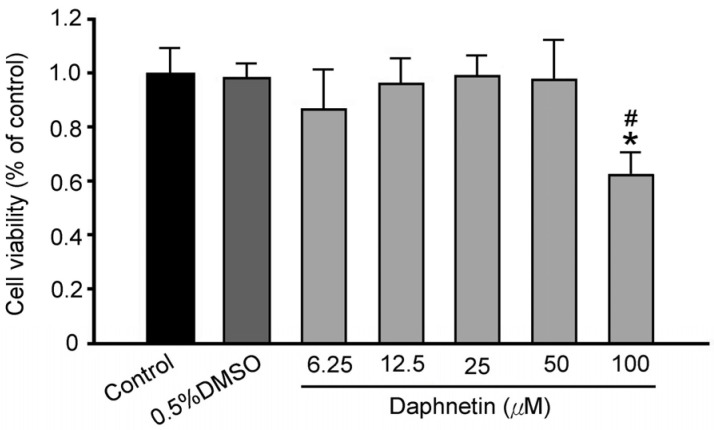
Effect of daphnetin concentration on cell viability of RSC96 Schwann cells. * *p* < 0.05 compared to the NG (5.5 mM glucose) group; ^#^ *p* < 0.05 compared to the NG + DMSO group.

**Figure 3 plants-13-03066-f003:**
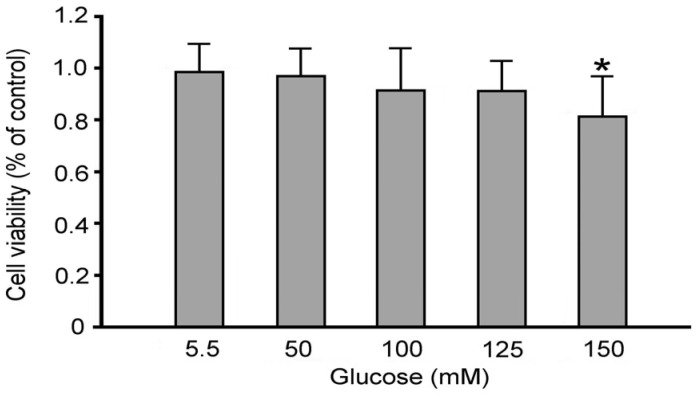
Effect of glucose concentration on cell viability of RSC96 Schwann cells. * *p* < 0.05 as compared to the NG (5.5 mM glucose) group.

**Figure 4 plants-13-03066-f004:**
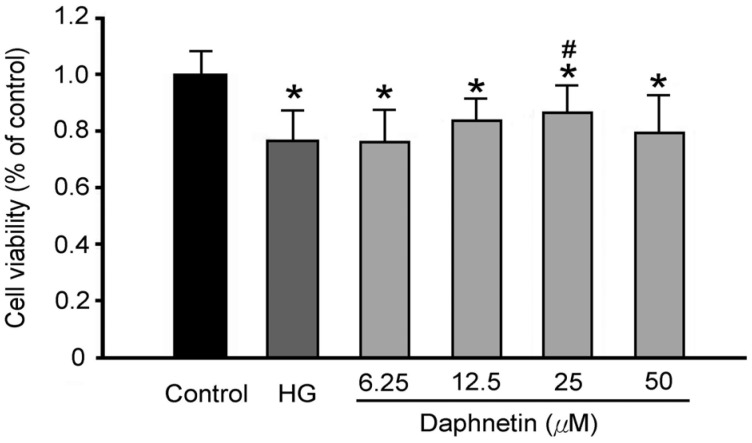
Effect of daphnetin concentration on cell viability of RSC96 Schwann cells incubated in a high-glucose medium. * *p* < 0.05 compared to the NG (5.5 mM glucose) group; ^#^ *p* < 0.05 compared to the HG (150 mM glucose) group.

**Figure 5 plants-13-03066-f005:**
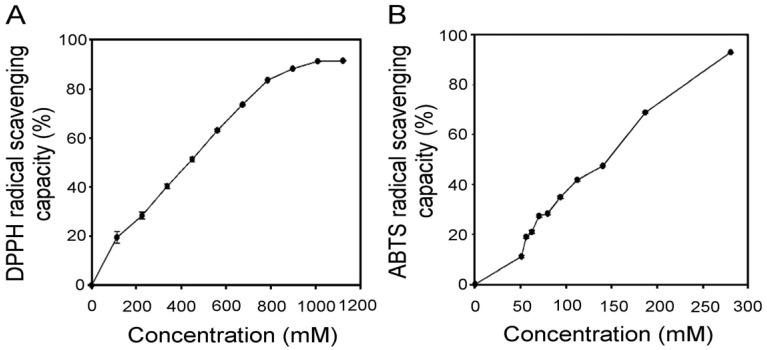
Effects of daphnetin on radical scavenging capacity of (**A**) DPPH and (**B**) ABTS in RSC96 Schwann cells incubated in a high-glucose medium. Data are expressed as mean ± SD from triplicate experiments.

**Figure 6 plants-13-03066-f006:**
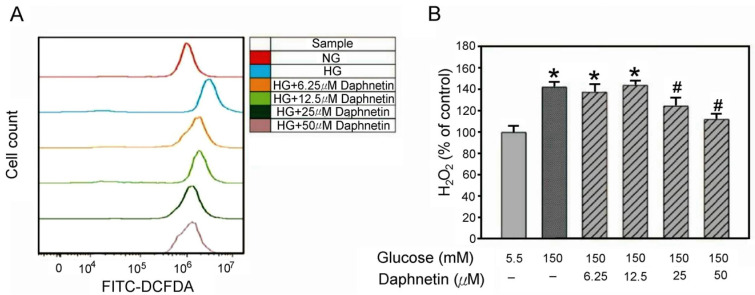
Effect of daphnetin on H_2_O_2_ levels in RSC96 cells incubated in a high-glucose medium. Fluorescence intensity measured by flow cytometry (**A**) and Quantitative analysis of fluorescence intensity (**B**). * *p* < 0.05 compared to the NG (5.5 mM glucose) group; ^#^ *p* < 0.05 compared to the HG (150 mM glucose) group.

**Figure 7 plants-13-03066-f007:**
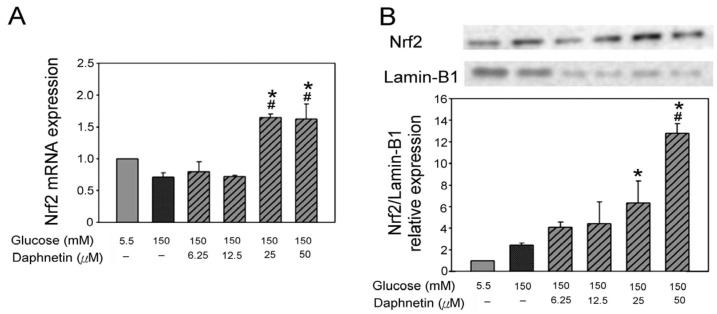
Nrf2 mRNA expression (**A**) and nuclear Nrf2 protein expression (**B**) in RSC96 cells incubated in a high-glucose medium with daphnetin. * *p* < 0.05 compared to the NG (5.5 mM glucose) group; ^#^ *p* < 0.05 compared to the HG (150 mM glucose) group.

**Figure 8 plants-13-03066-f008:**
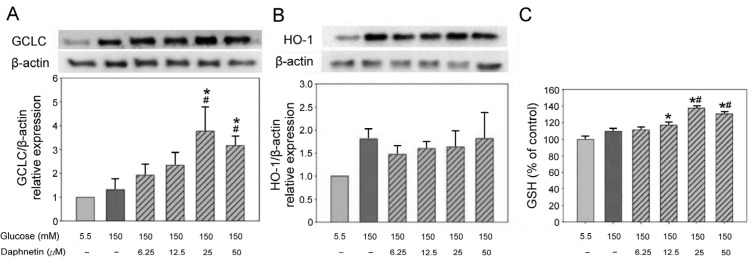
Protein expression of GCLC (**A**), HO-1 (**B**) and GSH content (**C**) in RSC96 Schwann cells incubated in a high-glucose medium with daphnetin. * *p* < 0.05 compared to the NG (5.5 mM glucose) group; ^#^ *p* < 0.05 compared to the HG (150 mM glucose) group.

**Figure 9 plants-13-03066-f009:**
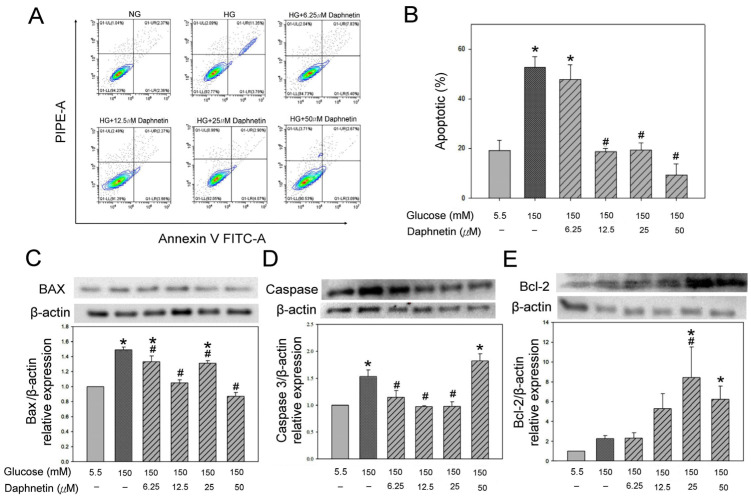
Fluorescence intensity measurement by flow cytometry (**A**), quantitative analysis (**B**), protein expression of Bax (**C**), Caspase 3 (**D**) and Bcl-2 (**E**) in RSC96 Schwann cells incubated in a high-glucose medium with daphnetin. * *p* < 0.05 compared to the NG (5.5 mM glucose) group; ^#^ *p* < 0.05 compared to the HG (150 mM glucose) group.

**Figure 10 plants-13-03066-f010:**
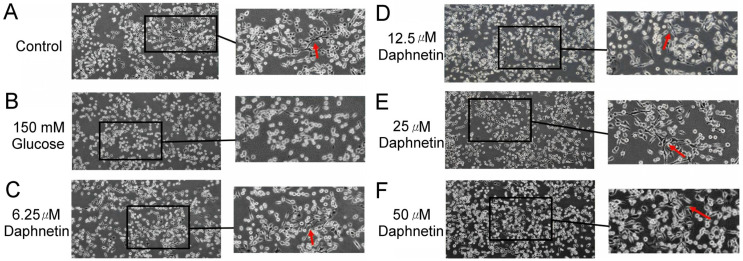
Effect of daphnetin on the morphology of RSC96 Schwann cells incubated in a high-glucose medium (**A**) control, (**B**) 150 mM glucose, (**C**–**F**) 6.25-50 μM daphnetin. (100× magnification, arrows denote spindle-shaped cells).

**Figure 11 plants-13-03066-f011:**
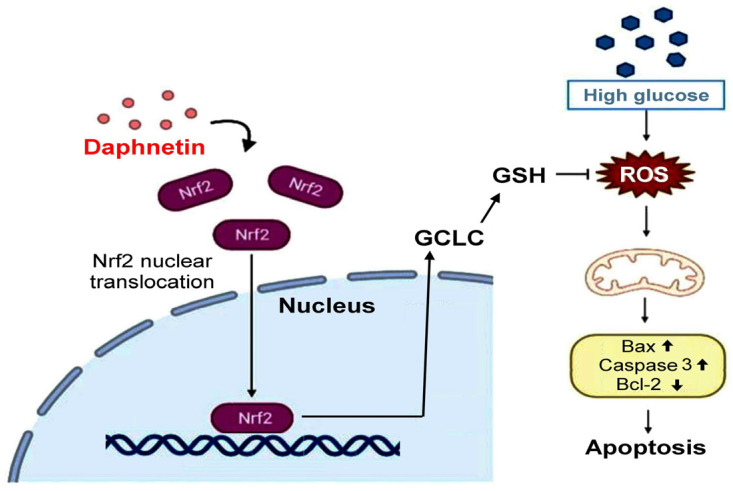
The possible mechanism of daphnetin in reducing high-glucose-induced Schwann cell apoptosis.

## Data Availability

Data are contained within the article.

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
