# Peer review of "Daphnetin Protects Schwann Cells Against High-Glucose-Induced Oxidative Injury by Modulating the Nuclear Factor Erythroid 2-Related Factor 2/Glutamate–Cysteine Ligase Catalytic Subunit Signaling Pathway"

_plants, 2024, doi:10.3390/plants13213066_

Round 1
Reviewer 1 Report
Comments and Suggestions for Authors
The paper ‘Daphnetin protects Schwann cells against high glucose-induced oxidative injury by modulating the Nrf2/GCLC Signalling Pathway’ deals with a very interesting topic, but has some shortcomings. Firstly, the introduction can certainly be improved and expanded, focusing more on the description of Schwann cells in neuropathy and diabetes and also on the involvement of oxidative stress in diabetic peripheral neuropathy. moreover, to be accepted, I think the work needs to be improved on other points:
· correct grammatical and spelling errors
· for dafnetin, a toxicity test was carried out in the absence of glucose to check the doses that could be used for the cells
· since this cell line was chosen, other cell lines can also be used to assess the response of other cells in a glucose-rich environment
· in the discussion, describe the results obtained on the basis of other studies in the literature
· update references with more recent work
Comments on the Quality of English LanguageImprove the quality of English and correct spelling and grammatical errors.
Author Response
Response letter to Reviewer 1 October 18, 2024
Dear Reviewer 1,
Appreciate so much for the efforts you have done to help us to improve the manuscript plants-3188675 with the title “Daphnetin protects Schwann cells against high glucose-induced oxidative injury by modulating the Nrf2/GCLC signaling pathway”. We realize a lot of extraordinary efforts you have done to help us to improve the manuscript. As requested, we re-wrote the manuscript carefully based on the suggestions from Reviewer 1 and the revised parts were red-labeled.
The following please see our replies to your comments point by point.
Best Regards,
Szu-Chuan Shen Ph. D.
Reviewer 1 Comments:
- The paper ‘Daphnetin protects Schwann cells against high glucose-induced oxidative injury by modulating the Nrf2/GCLC Signaling Pathway’ deals with a very interesting topic, but has some shortcomings. Firstly, the introduction can certainly be improved and expanded, focusing more on the description of Schwann cells in neuropathy and diabetes and also on the involvement of oxidative stress in diabetic peripheral neuropathy.
Reply:
Accept. We addressed the relevant descriptions in the Introduction section accordingly (lines 49-59).
- Correct grammatical and spelling errors.
Reply:
Thank you for your comment. The manuscript has been thoroughly revised by our native English-speaking colleagues to address all grammatical and spelling errors. The revised manuscript reflects these improvements to enhance clarity and readability.
- For dafnetin, a toxicity test was carried out in the absence of glucose to check the doses that could be used for the cells.
Reply:
Thank you for your comment on the daphnetin toxicity test. Recognizing that RSC96 Schwann cells require glucose for survival, we maintained a control with 5.5 mM glucose to mimic physiological conditions (revised Figure 2). This approach allowed us to accurately assess daphnetin's safety and distinguish its cytotoxic effects from its protective role against high-glucose-induced cell death (revised Figure 4).
- Since this cell line was chosen, other cell lines can also be used to assess the response of other cells in a glucose-rich environment.
Reply:
Thanks for the great suggestion. We chose the RSC96 Schwann cell line because it is a well-established model for studying diabetic peripheral neuropathy (DPN) and Schwann cell dysfunction under hyperglycemic conditions. We agree that future studies could extend these findings by examining other cell lines to provide a broader understanding of cellular responses in diabetic conditions. The relevant description has been also added in the revised manuscript (lines 244-247).
- In the discussion, describe the results obtained on the basis of other studies in the literature.
Reply:
Accept. The relevant results obtained by other studies in the literature have been addressed in the Discussion section accordingly (lines 195-235).
- Update references with more recent work.
Reply:
Agree. Relevant literature from the past five years has been reviewed, updated, and addressed into the manuscript to provide a more comprehensive and current context for our findings (lines 369-371, 419-420).
Reviewer 2 Report
Comments and Suggestions for Authors
Article
Daphnetin protects Schwann cells against high glucose-in-duced oxidative injury by modulating the Nrf2/GCLC Signal-ing Pathway
A brief summary
The research was designed in accordance with the current state of knowledge, the work was conducted in an appropriate way, the results are presented in a straightforward and unambiguous manner, and the conclusions provide a comprehensive summary of the findings. Just a few minor, detailed comments that could possibly be taken into account.
Broad comments
1. The present study aimed to investigate the effect of daphnetin, a coumarin derivative, on the viability of RSC96 Schwann cells under high glucose conditions. Furthermore, the antioxidant capacity was evaluated using two assays: DPPH and ABTS. Additionally, the effects of oxidative stress markers, such as Hâ‚‚Oâ‚‚, and the regulation of Nrf2 expression were investigated.
2. The principal strength of this research is the extensive range of work proposed and properly carried out by the authors.
3. The study was able to confirm the action of one of the coumarins that are naturally synthesised by plants and propose an explanation for this process. This could represent a significant advance in the development of efficacious pharmaceuticals derived from natural products, while simultaneously reinforcing the position of phytotherapy.
4. A methodology was proposed that permitted a comprehensive examination of the material and verification of the theses set forth in this research. It is important to note that the methodology is presented in a straightforward and accessible manner.
5. The conclusions are in alignment with the presented results and provide a comprehensive summary thereof. All issues raised are discussed with specific reference to the relevant experiments.
6. The references are appropriate.
7. Additional comments and suggestions can be found below.
Specific comments
Line 40 'Keywords' is an excellent place to insert phrases about ongoing research that is not included in the title. E.g. 'RSC96 cells', 'Bcl-2', 'Bax', 'apoptosis'.
Line 74. It would be beneficial to gain source or insight into the methodology employed to ascertain that the concentrations of daphnetin in question represent the maximum non-toxic concentrations.
Figure 4. Please indicate the number of replicates (n) that were performed. At the initial value, the graph (A) displays a slight whisker of standard deviation. Please explain why graph (B) appears to be so distorted.
Figure 10. The chemical formula is probably unnecessary here, however. It is better to put it in the text, if it already has to stay in the article.
Author Response
Response letter to Reviewer 2 October 18, 2024
Dear Reviewer 2,
Appreciate so much for the efforts you have done to help us to improve the manuscript plants-3188675 with the title “Daphnetin protects Schwann cells against high glucose-induced oxidative injury by modulating the Nrf2/GCLC signaling pathway”. We realize a lot of extraordinary efforts you have done to help us to improve the manuscript. As requested, we re-wrote the manuscript carefully based on the suggestions from Reviewer 2 and the revised parts were red-labeled.
The following please see our replies to your comments point by point.
Best Regards,
Szu-Chuan Shen Ph. D.
Reviewer 2 Comments:
- Line 40 'Keywords' is an excellent place to insert phrases about ongoing research that is not included in the title. E.g. 'RSC96 cells', 'Bcl-2', 'Bax', 'apoptosis'.
Reply:
Agree. The terms include RSC96 cells, Bcl-2, Bax, and apoptosis were added in the Keywords of revised manuscript (lines 39-40).
- Line 74. It would be beneficial to gain source or insight into the methodology employed to ascertain that the concentrations of daphnetin in question represent the maximum non-toxic concentrations.
Reply:
Thank you for the great comment. In this study, the MTT assay was used to determine the maximum non-toxic concentrations of daphnetin on Schwann cells. Our results indicated that daphnetin was non-toxic at concentrations ranging from 6.25 μM to 50 μM (revised Figure 2, lines 79-80), providing a basis for its use in subsequent experiments.
- Figure 4. Please indicate the number of replicates (n) that were performed. At the initial value, the graph (A) displays a of standard deviation. Please explain why graph (B) appears to be so distorted.
Reply:
Thank you for the suggestion. We confirm that each experiment was performed in triplicate (n=3). The foot note of the revised Figure 5 was revised accordingly (lines 112-113). Regarding to the initial value that slight whisker in graph (A) and distort observed in graph (B), the shape of the curve reflects the actual data obtained from the experiments, which were thoroughly repeated to confirm consistency. This non-linear response may be due to the biological variability of the Schwann cells under the experimental conditions.
- Figure 10. The chemical formula is probably unnecessary here, however. It is better to put it in the text, if it already has to stay in the article.
Reply:
Agree. The original Figure 10 was moved to the Introduction section as the revised Figure 1 in the revised manuscript. The number of all figures in the manuscript was also revised accordingly.
Reviewer 3 Report
Comments and Suggestions for Authors
The manuscript presents important information although it does not represent a real model of diabetic neuropathy. It does not include limitations of the study. It also does not emphasize the osmotic effect of glucose in the cell assay.
According to Figure 1, the most appropriate dose should be 12.5 or 25 microM since 50 is closer to its toxic dose of 100 microM.
In Figure 3, the only dose with a significant difference compared to the HG group is 25 microM. Why does it mention in line 75 that 50 microM was used in the tests? This is not consistent.
Figure 1 mentions NG as 25 mM glucose but on line 80 and Figure 2 NG = 5.5 mM glucose. Please review that information.
In Figure 2 and throughout the manuscript, the effect of oxidative stress is mentioned as responsible for the decrease in cell viability, but the osmotic effect of hyperglycemia is never mentioned. Mannitol could be an important osmotic control in the model. There is no discussion on this.
Line 100. The IC50 of 435.4 micro M is much higher than that used in the study (50 micro M). At this dose the antioxidant effect should be very limited. There is no discussion on this.
Very different from the ABTS assay where it mentions an IC50 of 26 micro M, although that value does not agree with Figure 4B. Please Review
Figure 4 mentions mM but the other figures mention microM. Please check the concentration.
Author Response
Response letter to Reviewer 3 October 18, 2024
Dear Reviewer 3,
Appreciate so much for the efforts you have done to help us to improve the manuscript plants-3188675 with the title “Daphnetin protects Schwann cells against high glucose-induced oxidative injury by modulating the Nrf2/GCLC signaling pathway”. We realize a lot of extraordinary efforts you have done to help us to improve the manuscript. As requested, we re-wrote the manuscript carefully based on the suggestions from Reviewer 3 and the revised parts were red-labeled.
The following please see our replies to your comments point by point.
Best Regards,
Szu-Chuan Shen Ph. D.
Reviewer 3 Comments:
- The manuscript presents important information although it does not represent a real model of diabetic neuropathy. It does not include limitations of the study. It also does not emphasize the osmotic effect of glucose in the cell assay.
Reply:
Thank you for the great suggestion. The limitations of this study were addressed in the Discussion section of revised manuscript (lines 243-249). In addition, regarding to the osmotic effect of glucose in this study, we used a medium contains 150 mM glucose that ensures the osmolarity remained within the range of 260–320 mOsm/kg and does not affect the normal growth of Schwann cells in this high glucose condition (Yang et al., 2016; Kusano et al., 1999).
References
Yang, X.; Yao, W.; Shi, H.; Liu, H.; Li, Y.; Gao, Y.; Liu, R.; Xu, L. Paeoniflorin protects Schwann cells against high glucose induced oxidative injury by activating Nrf2/ARE pathway and inhibiting apoptosis. J. Ethnopharmacol. 2016, 185, 361-9.
Kusano, K., House, S. B., & Gainer, H. (1999). Effects of osmotic pressure and brain-derived neurotrophic factor on the survival of postnatal hypothalamic oxytocinergic and vasopressinergic neurons in dissociated cell culture. Journal of neuroendocrinology, 11(2), 145–152.
- According to Figure 1, the most appropriate dose should be 12.5 or 25 microM since 50 is closer to its toxic dose of 100 microM.
Reply:
Thank you for the suggestion. According to our experimental results, we determined that 50 μM is the maximum non-toxic concentration for RSC96 Schwann cells in this study (lines 79-80). While 12.5 μM and 25 μM also fall within the non-toxic range, therefore we chose 50 μM to explore the full therapeutic potential of Daphnetin at its highest safe concentration in high-glucose conditions.
- In Figure 3, the only dose with a significant difference compared to the HG group is 25 microM. Why does it mention in line 75 that 50 microM was used in the tests? This is not consistent.
Reply:
Thank you for the suggestion. Our study indicated that under normal culture conditions (ie. 5.5 mM glucose), the maximum non-toxic concentration of daphnetin is 50 μM in revised Figure 2. However, the revised Figure 4 (original Figure 3) specifically analyzes the effect of daphnetin on the viability of RSC96 Schwann cells in a high-glucose environment (ie. 150 mM glucose), with 25 μM daphnetin being the most effective concentration.
- Figure 1 mentions NG as 25 mM glucose but on line 80 and Figure 2 NG = 5.5 mM glucose. Please review that information.
Reply:
Thank you for pointing out this type error. We correct the NG as 5.5 mM glucose in foot note of revised Figure 2 in the manuscript accordingly (line 92).
- In Figure 2 and throughout the manuscript, the effect of oxidative stress is mentioned as responsible for the decrease in cell viability, but the osmotic effect of hyperglycemia is never mentioned. Mannitol could be an important osmotic control in the model. There is no discussion on this.
Reply:
Thank you for your suggestion. Please refer the reply of question 1.
- Line 100. The IC50 of 435.4 micro M is much higher than that used in the study (50 micro M). At this dose the antioxidant effect should be very limited. There is no discussion on this.
Reply:
Accept. The unit and level of DPPH and ABTS IC50 value in revised Figure 5 have been corrected in the manuscript accordingly (lines 106, 108). The antioxidants effect of daphnetin, such as hydrogen peroxide content, on RSC96 Schwann cells under high glucose conditions was discussed in the manuscript (lines 206-211).
- Very different from the ABTS assay where it mentions an IC50 of 26 micro M, although that value does not agree with Figure 4B. Please Review
Reply:
Accept. The unit and level of ABTS IC50 value in revised Figure 5B have been corrected in the manuscript accordingly (line 108).
- Figure 4 mentions mM but the other figures mention microM. Please check the concentration.
Reply:
Accept. The unit and level of DPPH and ABTS IC50 value in revised Figure 5 have been corrected in the manuscript accordingly (lines 106, 108).
Round 2
Reviewer 3 Report
Comments and Suggestions for Authors
The attention to the suggested modifications in the previous review is appreciated, which undoubtedly improved the quality of the manuscript. Thank you!